Perception, knowledge, and attitude of medical doctors in Saudi Arabia about the role of physiotherapists in vestibular rehabilitation: a cross-sectional survey

Alyahya Danah
Kashoo Faizan Z. f.kashoo@mu.edu.sa
Department of Physical Therapy and Health Rehabilitation, College of Applied Medical Sciences , Al Majmaah , Riyadh , Saudi Arabia
Keogh Justin
Electronic publication date: 2022 Mar 7
Publication date: 2022
Volume: 10
Electronic Location ID: e13035
Received 2021 Oct 18; Accepted 2022 Feb 8
Copyright: ©2022 Alyahya and Kashoo
Copyright year: 2022
Copyright holder: Alyahya and Kashoo
License: This is an open access article distributed under the terms of the Creative Commons Attribution License, which permits unrestricted use, distribution, reproduction and adaptation in any medium and for any purpose provided that it is properly attributed. For attribution, the original author(s), title, publication source (PeerJ) and either DOI or URL of the article must be cited.
License URL: https://creativecommons.org/licenses/by/4.0/

Keywords: Vestibular rehabilitation, Referral, Physical therapy modalities, Evidence-based practice, Vestibular disorders, Balance disorders, Dizziness, Cross-sectional survey

Funding: The Deanship of Scientific Research at Majmaah University R-2022-28 The Deanship of Scientific Research at Majmaah University supported this work under Project Number No. R-2022-28. The funders had no role in study design, data collection and analysis, decision to publish, or preparation of the manuscript.

==============================
Objectives

There is compelling scientific evidence about the role of physiotherapists in vestibular rehabilitation. However, patients with vestibular-associated dizziness and balance disturbances are seldom referred to physiotherapists in Saudi Arabia. Therefore, this study aims to achieve insight into perceptions, knowledge, attitudes, and referral practices among Saudi Arabian medical doctors relating to the role of physiotherapists in vestibular rehabilitation.

Methods

A sample of 381 medical doctors practicing in Saudi Arabia participated in this nationwide cross-sectional study. The sample was obtained from 226 hospitals across 13 provinces of Saudi Arabia by stratified sampling method. The 23-item questionnaire developed by a team of experts was emailed to medical doctors practicing in various hospitals across Saudi Arabia.

Results

Out of 1,231 medical doctors invited, 381 medical doctors responded, giving a response rate of 30.9%. One hundred ninety-three (50.6%) medical doctors reported managing patients with vestibular rehabilitation. The most preferred specialist for managing patients with vestibular disorders was an Ear Nose Throat (ENT) specialist (n = 173, 89.6%). Related Sample Cochran’s Q test showed statistically significant difference between preferred specialist for managing patients with vestibular disorders (ENT specialists, physiotherapists, nurses, occupational therapists and audiologists) (χ2(4) = 482.476, p = 0.001). Out of 193 medical doctors, 153 (79.2%) reported no role of the physiotherapist in vestibular rehabilitation. One hundred forty-five (75.1%) of medical doctors reported that they were not aware of the role of physiotherapists in vestibular rehabilitation. Only 27 (15.5%) medical doctors reported referring patients with vestibular disorders to physiotherapists.

Conclusion

The study reports that physiotherapy services are underutilized in vestibular rehabilitation due to limited referral from Saudi Arabian medical doctors. Therefore, there is a need to increase the awareness among Saudi Arabian doctors about the physiotherapist’s role in vestibular rehabilitation.

Introduction

Physiotherapists globally use various therapeutic approaches to treat patients with dizziness and balance dysfunction due to vestibular disorders (Alghadir & Anwer, 2018). Vestibular rehabilitation (VR) is reported to be effective in vestibular disorders (Eleftheriadou, Skalidi & Velegrakis, 2012). In the last few years, there has been compelling evidence about the effectiveness of vestibular rehabilitation in the treatment of vestibular hypofunction (Porciuncula, Johnson & Glickman, 2012), benign paroxysmal positional vertigo (BPPV) (Rodrigues et al., 2018), persistent postural-perceptual dizziness (PPPD) (Eldøen et al., 2021), vestibular migraine (Power et al., 2018), dizziness associated with multiple sclerosis (MS) (García-Muñoz et al., 2020), and vestibular involvement in traumatic brain injury (Kleffelgaard et al., 2019).

In the last few decades, there has been a steady growth in the number of colleges and universities across the Kingdom of Saudi Arabia. The private and government colleges in Saudi Arabia are constantly monitored by the National Commission for Academic Accreditation and Assessment (NCAAA) to compete with the international educational standards (Albaqami, 2019). To get accredited by NCAAA, colleges and universities in Saudi Arabia strive to achieve the highest educational standards (AlShammery, 2017). These policies and procedures led to changes in the academic curriculum of various disciplines, especially the physical therapy profession. Physical therapy education in Saudi Arabia is changing from rigid curricula-based education to more flexible problem-based learning. Physical Therapists (PTs) use advanced technology such as virtual reality and optokinetic scenes to improve balance and alleviate dizziness in vestibular disorders (Alahmari, 2012; Bergeron, Lortie & Guitton, 2015). In addition, the introduction of evidence-based practice enables emerging physical therapists to treat the vast majority of vestibular disorders (Alghadir, Iqbal & Whitney, 2013; Arnold et al., 2017; Hall et al., 2016a; Klatt et al., 2015). A research article published in 2009 reported the recommendation by the Barany Society Ad Hoc Committee on vestibular rehabilitation therapy from 113 physiotherapists from 19 countries (Cohen et al., 2009). The committee recommended the inclusion of evidence-based vestibular rehabilitation in education and practice. Similarly, a study conducted in Europe reported that 90% of postgraduate students undergo training in VR (Meldrum et al., 2020). Moreover, licensed physical therapists in Saudi Arabia participate in workshops and hands-on training in various advanced techniques as well as vestibular rehabilitation. These trainings are mandatory to earn credit hours and retain the license to practice in Saudi Arabia. Most of the patients with vestibular disorder initially visit a physician before being referred to a physiotherapist. However, advanced skills learned during academic years are underutilized due to non-referral, which may not be restricted to patients with vestibular disorders. Therefore, it is important to understand the perceptions, knowledges, and attitudes surrounding medical doctors’ referral of patients to a physical therapist for VR.

Method

Study design

The study was conducted based on the STROBE checklist for cross-sectional studies (Appendix S1). Thirteen-hundred members of the Saudi Commission for Health Specialties (SCFHS) with different specialties, designations, nationalities, and work experience, working in various hospitals in 13 regions of Saudi Arabia, were invited to participate in this study. Medical doctors were invited through email to participate in the survey. The study commenced in April 2021 and completed in September 2021. The study was approved by the Institutional Review Board of Majmaah University (MUREC-April. 7/COM-2021/31-1). Medical doctors received an email that contained a link to the survey and a consent form. Additionally, human resource departments of hospitals for contacted to distribute the survey among medical doctors.

Participants

All licensed medical doctors working in KSA were entitled to take part in the survey. The participants did not receive any incentives for participation. Email addresses of the participants were retrieved from the hospital website or by contacting the human resource department of the hospital. Participants received informed consent forms through email, which included all survey-related information as well as contact information of the correponding author. For two months, non-responsive participants were reminded every two weeks. Further communication was stopped, if the participants did not answer after four reminders.

Sampling method

Stratified sampling was used to uniformly collect data from medical doctors working in 13 provinces of Saudi Arabia. There are 226 hospitals recognized by Ministry of Health Saudi Arabia, with a vast difference in the number of hospitals in each of the 13 provinces. Therefore, a proportionate sampling method was employed to obtain a homogenous sample depending on the number of hospitals in each province. To obtain an optimal sample size of 371 medical doctors, 10% of hospitals were randomly selected from each province. For example, four hospitals were randomly picked from 48 hospitals in Riyadh province, accounting for 10% of the total hospitals in the province. A total of 23 hospitals were randomly selected from a computer-generated random number assigned to 226 hospitals in 13 provinces across the country. An official invitation letter was sent to each hospital’s human resource department to disseminate the survey among medical doctors. A total sample size of 1,231 medical doctors were invited to participate, and 381 medical doctors responded, giving a response rate of 30.9%

Survey questionnaire development

The questionnaire was developed by a team of experts in vestibular management (6 medical doctors, one language expert, and 2 senior physiotherapists) and piloted among ten medical doctors before distribution. Cronbach’s alpha for the reliability of the questionnaire was 0.843. The survey questions were inserted into the Qualtrics survey software, and the questionnaire link was provided to medical doctors through email.

The survey comprised seven sections and 23 questions; the first section contained ten general items (questions 1–10), the second section comprised questions related to vestibular patients (questions 11–13), and the third section consisted of questions related to the attitudes and knowledge of medical doctors regarding physical therapists’ role in vestibular disorders (questions 14–18). if the medical doctors answered that they did not see the patient with vestibular patients, the questionnaire was terminated after collecting demographic information. The fourth section consisted of questions related to referral (question no. 19). The fifth section consisted of one question related to the involvement of physiotherapists in the management of different vestibular conditions (question no. 20). The sixth section consisted of questions related to the perception of medical doctors about assessment and maneuvers effectively applied by a physiotherapist (question no. 21–22). Finally, the seventh section consisted of one question related to feedback obtained from medical doctors about the approach of a physiotherapist with patients with vestibular disorders (question no. 23) (Appendix S2).

Sample size calculation

The sample size was determined from the online Raisoft sample size calculator (Inc, 2020) and based on the response rate of 50%, a confidence interval of 95%, and a margin of error of 5%, with a total medical doctors population of around 113,146 as per the 2019 census (Health, 2019). Therefore, the sample size required was 377.

Statistical analysis

IBM SPSS version 20 (IBM Corp., Armonk, NY, USA) was used to analyze the data. Qualitative data were analyzed as frequencies and percentages, nominal data by binomial test, and multiple responses by chi-square test at a threshold value of 50%. Logistic regression was used to estimate the effect of nationality, experience, specialty, and workplace on the referral practices and perceptions of doctors about the role of the physiotherapist in vestibular rehabilitation. A P < 0.05 was considered statistically significant.

Figure 1 Flow diagram of sampling and sample size calculation from 13 provinces of Saudi Arabia.

Figure 2 Nationality of the medical doctors in the study (n = 381).

Figure 3 Specialty of the sample medical doctors in the study (n = 381).

Results

Demographic data of medical doctors

A total of 381 medical doctors participated in the study, with a response rate of 30.9% (Fig. 1). The data were collected from 381 medical doctors, with nearly equal participation from male (n = 187, 49.15%) and female medical doctors. The highest participation came from the 24–30-year-old age group (n = 111, 29.1%) and the lowest from the 60 years and over age group (n = 61, 16%). The majority of the medical doctors were native Saudi citizens (n = 158, 41.5%), and those from Canada or America made up the smallest percentage (n = 1, 0.3%) (Fig. 2). The most common specialization was dentistry (n = 90), followed by ENT (Ear Nose Throat) (n = 85), and the least common was gastroenterology (n = 1) (Fig. 3). The highest number of medical doctors reported that they obtained their last degree from Saudi Arabia (n = 119, 31%) and the least from the Philippines (n = 1, 0.9%) (Fig. 4). The highest number of participants were from Riyadh province (n = 122, 32%) and the least from the northern border (n = 13, 3.4%) and Asir province (n = 13, 3.4%) of Saudi Arabia (Fig. 5). The majority of medical doctors were practicing at a government hospital (n = 255, 66.9%). One hundred ninety-three (50.7%) medical doctors reported managing and treating the patients with vestibular disorders, and the plurality (n = 154, 40.4%) of them reported managing and treating less than five patients in a month (Table 1). The common vestibular condition seen by medical doctors was benign paroxysmal positional vertigo (n = 169, 87.6%), and the least common was vestibular migraine (n = 33, 17.1%) and mal de barquement (n = 33, 17.1%) (Fig. 6).

Figure 4 Country of last medical education among medical doctors (n = 381).

Figure 5 Medical doctors from 13 provinces of Saudi Arabia.

The attitude of 193 medical doctors who were involved in management of patients with vestibular conditions

Medical doctors reported that the most preferred medical professionals for managing patients with the vestibular disorder were ENT specialists (n = 173, 72.1%), followed by physiotherapists (n = 30, 12.5%), and the least preferred were nurses (n = 12, 5.0%) (Table 2). Friedman’s two-way analysis of variance by ranks showed a statistically significant difference in the proportion of medical doctors reporting preferred healthcare professionals for the management of patients with vestibular disorders (ENT, Audiologist, nurse, PT, OT), χ2(4) = 482.476, p = 0.001. The pairwise comparison Bonferroni alpha adjustment between health professionals revealed a significant difference between OT vs. ENT (p = 0.001), Nurses vs. ENT (p = 0.001), Audiologist vs. ENT (p = 0.001), and PT vs. ENT (p = 0.001).

Table 1 Demographic data and participants characteristics.

Variable name	Grouping	Frequency	Percentage	P	
Gender	Male	187	49.1	0.759*	
Female	194	50.9	
Age groups	24-30 years	111	29.1	0.001**	
31–40 years	71	18.6	
41–50 years	105	27.6	
51–60 years	72	18.9	
60 above	22	5.8	
Experience as a medical doctor	1–3 years	107	28.1	0.001**	
4–8 years	75	19.7	
9–15 years	138	36.2	
More than 15 years	61	16.0	
Work place	Government hospital	255	66.9	0.001**	
Private hospital	108	28.3	
Clinic	18	4.7	
Do you treat patients with vestibular disorders	yes	193	50.7	0.838*	
No	188	49.3	
How many patients with vestibular disorders do you treat in a month	Less than 5 patients	154	40.4	0.001**	
5–10 patients	33	8.7	
11–20 patients or more	6	1.6	
Notes.

* Binomial test (2-sided).

** One sample chi-square test.

Knowledge of 193 medical doctors about the role of the physiotherapist in vestibular assessment, diagnosis, and treatment

Out of 193 medical doctors, 163 (84.5%) responded to the question related to role of physiotherapist in assessment and treatment and remainder 30 (15.5%) responded in favor of physiotherapist. Out of 177 mutually inclusive responses from 163 respondents, (n = 156, 80.8%) medical doctors were not aware of the role of the physiotherapist in the assessment and diagnosis of a patient with vestibular disorders. The remainder of the 21 responses from the medical doctors stated that a lack of confidence (n = 10, 5.2%), lack of experience (n = 6, 3.1%), and insufficient knowledge among physiotherapists (n = 5, 2.6%) were the primary barriers. Similarly, out of 193 medical doctors, 155 (80.3%) responded to the question related to role of physiotherapist in treatment of patient with vestibular rehabilitation, remainder 30 (15.5%) responded in favor of physiotherapist and 8 (8%) choose not to respond to the question due to unknown reason. Out of 156 mutually inclusive responses from 155 medical doctors (n = 145, 75.1%) were not aware of the role of the physiotherapist in the treatment of a patient with vestibular disorders and the remainder reported less experience (n = 9, 4.7%) and insufficient knowledge among physiotherapists (n = 2, 1.0%) (File S1).

Figure 6 Common vestibular cases seen by medical doctors (n = 193).

Logistic regression was performed to estimate the effects of nationality, specialty, experience, and workplace on the likelihood that medical doctors reported that physiotherapists are capable of assessing, diagnosing, and treating patients with vestibular disorders. The logistic regression model was statistically significant, χ2(30) = 76.343, p = 0.001. The model explained 56.5% (Nagelkerke R2) of the variance among medical doctors and correctly classified 90.7% of cases. Medical doctors with 1–5 years of experience are 7.165 (95% CI [1.104–46.515]) times more likely to report that physiotherapists are involved in assessing, diagnosing, and treating patients with vestibular disorders than medical doctors with more experience. The other predictors such as nationality, specialty, and workplace were not significant.

Out of 193 medical doctors, 30 reported that physiotherapists could effectively apply assessment techniques in patients with vestibular disorders. The most frequently reported assessment technique that physiotherapists could effectively apply was the Dix Hill Pike maneuvers (n = 23, 11.9%), followed by the Dynamic Visual Acuity Test (n = 21, 10.9%), and the least was VOR-Cancellation (n = 16, 8.3%).

Out of 193 medical doctors, 30 reported that physiotherapists can effectively use maneuvers and exercises in patients with vestibular disorders. The most frequently reported maneuver that physiotherapists can effectively apply was the Semont maneuver (21, 10.9%), followed by the Canalith Repositioning maneuver (n = 20, 10.4%), and the least was the Barbecue roll maneuver (n = 11, 5.7%).

Perception of 193 medical doctors about the role of physiotherapist in different vestibular conditions

The majority of medical doctors reported no role of physiotherapists (78.4%), and merely 2.0% and 2.7% reported a role in assessment and treatment in common vestibular conditions, respectively. The majority of medical doctors reported no role of the physiotherapist in BPPV (78.8%), vestibular neuritis (87.6%), vestibular hypotension (80.3%), vestibular migraine (86%), Meniere’s disease (83.9%), multiple sclerosis (86.5%), mal de debarquement (1%), perilymphatic fistula (100%), cervicogenic dizziness (93.8%), vestibular involvement due to traumatic brain injury (94.3%), and motion sensitivity (86.0%). A relatively small number of medical doctors reported that physiotherapists are involved in assessment (0.5 to 4.1%) and treatment (0.5 to 13%) of patients’ vestibular conditions. BPPV was the most frequent vestibular condition in the assessment and treatment of which physiotherapists were thought to have a role.

Referral attitude of medical doctors

Out of 193, only 27 (14.0%) medical doctors referred vestibular cases to a physiotherapist. The majority of cases referred to physiotherapists were benign paroxysmal vertigo (23, 20%), and the least common types of cases were vestibular migraine (n = 1, 0.9%) and perilymphatic fistula (n = 1, 0.9%) (Table 3). Out of 27 referrals to physiotherapists, the highest number of referrals to physiotherapists were from Saudi national medical doctors (n = 16, 45.7%), followed by Indian medical doctors (n = 10, 28.6%). The majority of referrals to physiotherapist were made by general physicians (n = 11, 40.7), followed by ENT specialists (n = 5, 18.5%) (Fig. 7).

Feedback from the patient after treatment from the physiotherapist

The majority of medical doctors did not obtain feedback from patients (n = 168, 87%), and remainder reported their patients were moderately (n = 13, 6.7%) and highly (n = 12, 6.2%) satisfied with the treatment provided by the physiotherapists.

Discussion

In our study, 381 medical doctors participated; only 193 medical doctors reported managing patients with vestibular disorders. A significantly low number of medical doctors referred patients with vestibular disorders to physiotherapists. The majority of medical doctors reported no role of the physiotherapist in vestibular rehabilitation. The main reason for non-referral was that the medical doctors thought that the physiotherapist has no role in the management of patient with vestibular rehabilitation. A relatively smaller number of medical doctors reported that physiotherapists have less knowledge and experience to handle cases with vestibular rehabilitation.

Table 2 Who should manage patients with vestibular condition.

Statement	Ear nose throat specialist (n,%)	Physical therapist (n,%)	Occupational therapist (n,%)	Audiologist (n,%)	Nurse (n,%)	p	
Who should manage patients with vestibular conditions	173 (72.1%)	30 (12.5%)	5 (2.1%)	20 (8.3%)	12 (5%)	0.001*	
Notes.

* Related-samples Friedman’s two-way analysis of variance by ranks.

Table 3 The number of medical doctors referring the patients with vestibular disorders to a physiotherapist.

Vestibular conditions referred by medical doctors to physiotherapist (n = 27)	Frequency of referral for each condition	Percentage	
Benign paroxysmal positional vertigo	23	20.0	
Motion Sensitivity	20	17.4	
Vestibular symptoms associated with multiple Sclerosis	18	15.7	
Vestibular symptoms associated with traumatic brain injury	20	17.4	
Cervicogenic Dizziness	18	15.7	
Vestibular hypo function	6	5.2	
Vestibular neuritis	3	2.6	
Meniere’s disease	3	2.6	
Mal de Barquement	2	1.7	
Vestibular migraine	1	0.9	
Perilymphatic fistula	1	0.9	

Figure 7 Number of cases referred to physiotherpist by medical doctors (n = 27).

Our study found that patients with vestibular disorders are managed mainly by general physicians and ENT specialists. Medical doctors manage vestibular disorders with medications and vestibular maneuvers. However, several vestibular disorders are associated with dizziness and balance impairments which requires physical therapy interventions (Balci et al., 2013; Kundakci et al., 2018; Meldrum et al., 2012). Recent studies report that the recovery of a patient with vestibular dysfunction is slow and incomplete without physical therapy intervention particularly in patients with associated balance and dizziness symptoms (Han, Song & Kim, 2011; Humphriss et al., 2001). Exercises that progressively challenge balance are reported to be effective for patients with vestibular disorders. The physical therapy techniques benefit many different vestibular conditions such as patients with non-compensatory peripheral vestibular disorders (Alghadir, Iqbal & Whitney, 2013), BPPV (Haripriya et al., 2014), stable or mixed central lesion (Schneider et al., 2014), multifactorial balance abnormalities (Salzman, 2010), post-ablation surgeries (Enticott, O’leary & Briggs, 2005) (acoustic neuroma resection, labyrinthectomy).

A study conducted among ENT specialists reported the need to change the current diagnosis and treatment method of a patient with vertigo and dizziness (Weckel et al., 2020). Authors also reported that ENT specialists urged the incorporation of advanced medical technology to understand the mechanism of disorder for a better quality of care (Weckel et al., 2020). A number of research studies have examined the potential for modern physical therapy technology for the assessment and treatment of patients with vestibular rehabilitation such as virtual reality to improve gaze stability during head movements (Xie et al., 2021), smartphone application for guiding and monitoring exercises (Moral-Munoz et al., 2021), force plateform to measure postural sway (Song, 2019). Moreover, studies report that the management of patients with vestibular dysfunction with evidence-based medicine augmented by advanced therapeutic exercise led to better patient outcomes (Heffernan, Abdelmalek & Nunez, 2021; Xie et al., 2021).

Benign paroxysmal positional vertigo

Benign paroxysmal positional vertigo (BPPV) was the most preferred vestibular condition for referral to physiotherapists by medical doctors. However, the majority of medical doctors reported that they were not aware of the role of the physiotherapist in the management of patients with BPPV. Nevertheless, many researchers report recent advancements in the field management of BPPV by a physiotherapist. Similarly, a study conducted among 175 patients with BPPV reported that 79% of patients were cured within one week of treatment by the Semont maneuver and exercises for vestibular rehabilitation (Garcia, 2005). In addition, a recent study conducted of 314 patients with BPPV of the posterior canal found that 91% of cases showed significant improvement in two or fewer treatment sessions of Canalith repositioning maneuvers (Power, Murray & Szmulewicz, 2020). This implies that those patient with BPPV who adhere to exercise gets a additive benefits in symptoms.

In our study, we found a small percentage of medical doctors referring migraine-related dizziness to physiotherapists (Table 3). A study conducted in 2005 by Gottshall, Moore & Hoffer (2005) reported significant improvement among patients with idiopathic migraine-associated vertigo after vestibular rehabilitation therapy administered by the physiotherapist. The authors of the study also reported improvement in dizziness handicap, activities-specific balance confidence, dynamic gait, and computerized dynamic posturography measures among patients with migraine-associated dizziness. Medications used to prevent migraine- related dizziness are effectives but must be individualized based on age, side-effect, contra-indications and pre-existing conditions. However, exerises carry no side effects and are reported to improve the tolerance pain and quality of life in patients with migraine-related dizziness (Eggers, 2007).

Meniere’s disease is a chronic disorder of the inner ear resulting in balance and hearing impairment. In our study, few medical doctors reported that they were aware of the role of the physiotherapist in the management of Meniere’s disease. However, a study conducted in the UK among 360 participants with Meniere’s disease reported significant improvement among participants receiving vestibular rehabilitation and symptom control techniques as compared to the control group. The symptom control technique used in the study is a self-management technique to reduce the dizziness symptoms by habituation exercises and reduction of stress (Yardley & Kirby, 2006). Similarly, a study conducted among 26 patients with Meniere’s disease improved on posturography sensory organization test after vestibular physical therapy exercises (Gottshall et al., 2005). Therefore, physiotherapists can offer a useful and inexpensive self management techniques and exercises to patients with Meniere’s disease.

In our study, the majority of medical doctors reported that physiotherapists had no role in the management of a patient presenting with vestibular symptoms due to traumatic brain injury and multiple sclerosis. Vestibular symptoms such as dizziness and balance impairment associated with multiple sclerosis (Hebert et al., 2011; Lotfi et al., 2021) and traumatic brain injury have been reported in many studies (Akin et al., 2017; Fausti et al., 2009; Fife & Kalra, 2015; Gurley, Hujsak & Kelly, 2013). Dizziness and balance impairment associated with vestibular hypofunction (Hall et al., 2016b), vestibular neuritis (Strupp et al., 1998), mal de barquement (Cha, Deblieck & Wu, 2016), and perilymphatic fistula (Larem, Al Shawabkeh & Aljariri, 2021) can be effectively managed with various physical therapy modalities. A review study conducted in 2018 by Kundakci et al. reported that vestibular exercises are effective in patients with chronic dizziness (Kundakci et al., 2018). The evidence suggests that the physical therapy intervention complements the main course of treatment in patients with variety of vestibular disorders (Hall et al., 2016b). The physical therapy role in various disorders with associated vestibular symptoms would considerably reduce the medical cost. However, a robust research evidence is needed in vestibular rehabilitation about the reduction of medical cost because of addition of physical therapy intervention.

Referral and feedback from the patient

In our study, we found that only approximately one-third of medical doctors referred cases to a physiotherapist. A study conducted in Saudi Arabia of 108 physicians about their perception of physical therapy services reported that physiotherapists lack knowledge and skill to assess and treat patients (Alshehri et al., 2018). A study conducted among 282 patients with peripheral vestibular disease reported an average of late medical consultation after months or years from the onset of symptoms (Jáuregui-Renaud et al., 2013). A study conducted in the US reported low referral to physiotherapists for vestibular rehabilitation. However, retrospective medical records also found that there was a steady increase in PT referrals for vestibular disorders from 6.2% in 2006 to 12.9% in 2015 (Dunlap et al., 2020). However, a cohort study conducted among 2,374 patients with vertigo and dizziness reported that 61% used medication and 41.3% utilized PT services (Grill et al., 2014). The scope of physiotherapy in vestibular rehabilitation is limited in Saudi Arabia. There is a urgent need to inform medical doctors practicing in Saudi Arabia about the role of physiotherapists in vestibular rehabilitation. Moreover, telerahbilitation for patients with vestibular dysfunction would be a convenient addition during COVID-19 pandemic (Aloyuni et al., 2020).

Majority of medical doctors incolved in the management of patients with vestibular disorders were not aware about the physical therapist being trained in vestibular rehabilitation. The Saudi Arabian doctors may benefit not only from further education into benefits of physical therapy for their patients, but also behaviour change theories and approaches, so that they can better promote the physical therapy services to their patients. These approaches include education, training, and enablement (Geissler & Zeber, 2020). These interventions foster improved communication with other disciplines and enhance the management of vestibular patients by providing holistic and comprehensive care (Davis et al., 1999). A review article reported interactive and multifaceted continuous medical education programs and clinical decision support systems are beneficial in improving knowledge, utilization of resources, and enhancing patient outcome (Chauhan et al., 2017).

The majority of medical doctors did not obtain feedback from the patient after referral. A relatively small number of doctors reported that their patients were moderate to highly satisfied with the treatment provided by the physiotherapists.

Limitations

The cross-sectional study involved a relatively small population of medical doctors. However, the stratified sampling method ensured that the response was homogenously obtained across all 13 provinces in Saudi Arabia. The online questionnaire was a bit longer, which could create response bias or increase non-respondents due to the busy schedule of medical doctors. There was a relatively low response rate (30%) and approximately 50% of the medical doctors were not involved in the assessment and treatment of patients with vestibular disorders. These low response rates and relative equivalence in the medical doctors who are involved or not involved with patients with vestibular disorders, may affect the generalisation of these results to the population of Saudi Arabian doctors. However, the representative sample of medical doctors who participated in the study might be an unbiased reflection of the wider Saudi medical doctors. The unbiased nature of our results was further supported by the fact that data from medical doctors were uniformly collected from 13 provinces of Saudi Arabia with wide range of specialization and experience.

Conclusion

The study concludes that the physical therapy vestibular rehabilitation skills for vestibular disorders are underutilized due to non-referrals. The main reason for non-referral was the lack of awareness among medical doctors and medical doctors’ perception about the lack of experience and knowledge of vestibular rehabilitation among physical therapists. Therefore, the awareness program among medical doctors, especially ENT and general physicians, about the role of physiotherapy in vestibular rehabilitation would improve the number of referrals.

Supplemental Information

Appendix S1 STROBE-Checklist of items that should be included in reports of cross-sectional studies

Click here for additional data file.

Appendix S2 Questionnaire of the study

Click here for additional data file.

Supplemental Information 3 Table of mutually inclusive responses

Click here for additional data file.

Supplemental Information 4 Raw data

Click here for additional data file.

We thank all the medical doctors for participating in this study despite their busy schedules due to COVID -19 pandemic.

Additional Information and Declarations

Competing Interests

Author Contributions

Human Ethics

Ethics

Data Availability

The authors declare there are no competing interests.

Danah Alyahya conceived and designed the experiments, performed the experiments, analyzed the data, authored or reviewed drafts of the paper, and approved the final draft.

Faizan Z. Kashoo conceived and designed the experiments, performed the experiments, analyzed the data, prepared figures and/or tables, and approved the final draft.

The following information was supplied relating to ethical approvals (i.e., approving body and any reference numbers):

The Majmaah University granted the Ethical approval to carry out the study (Ethical Application Ref: MUREC-April. 7/COM-2021/31-1

The following information was supplied relating to ethical approvals (i.e., approving body and any reference numbers):

Deanship of Scientific Research at Majmaah University approved the study.

The following information was supplied regarding data availability:

The raw data is available in the Supplemental File.

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
