# Peer review of "Perception, knowledge, and attitude of medical doctors in Saudi Arabia about the role of physiotherapists in vestibular rehabilitation: a cross-sectional survey"

_PeerJ, doi:10.7717/peerj.13035_

## Round 0.1 · original submission · Major Revisions

While the authors have conducted a study that has some relevance to vestibular rehabilitation, particularly in Saudi Arabia and similar countries, the reviewers and I share many reservations about the current quality of the submitted manuscript that will require substantial revisions prior to this being more seriously considered for publication in PeerJ. I therefore suggest you pay particular attention to the last two reviewers' comments in your resubmission.

Reviewer 1 ·

Basic reporting

Dear Dr Keogh
Vestibular rehabilitation is an important field where physiotherapists have been playing an important role and this study addresses the awareness of the same amongst doctors.

Experimental design

The Study is well designed and well executed

Validity of the findings

I am concerned about few issues with this paper which the authors need to answer.
1. On what basis where the hospitals chosen? When you say randomly selected what do you mean ?
2. Why were dentists included in this study?
3. Ideally Vestibular Dysfunction cases are managed by ENT doctors, why was this study not restricted to ENT doctors ?
4. Out of the doctors recruited in this study only 50 % had the experience of managing Vestibular disease, in that case how will those who do not manage vestibular disease know about the role of a physiotherapist in vestibular disease ?

Additional comments

Overall the article is well writen.

Reviewer 2 ·

Basic reporting

There are many grammatical and typo errors, and unclear sentences. This study should be proofread and edited before publication.
Some of the reported findings in the result section were not founded in the tables/figures.
The discussion section needs to be reorganized, rearranged, and rewritten clearly.


TITLE
L1
The title should be rewritten correctly. It might be written as: “Perception, knowledge, attitude of medical doctors in Saudi Arabia about the role of physiotherapists in vestibular rehabilitation: A cross-sectional survey.”.

INTRODUCTION
L78
As I know that PTs are not required to study vestibular rehabilitation to retain license to practice physiotherapy, whether globally or at any other country. There is an important point in this paper that should be clarified which is that not all PTs are trained and have certificates in vestibular rehabilitation. Furthermore, not all clinics or hospitals have a certified PT in vestibular rehabilitation.

Experimental design

METHODS
L98
What about the specialty of doctors eligible to participate in the survey! What were their specialties? I see that there was a major limitation in this study that even doctors who are not specialized in vestibular or do not assess/treat those patients were also included in the study (n=188). So, how would they be involved in the study? I would see that excluding them because including them in the study will impact the results. If you already did that, please clarify it in the methods/results sections.

L118
Did the survey also included questions related to demographic factors?

L125-L134
Typo error! I think you mean “section” and not “session”.

L132
The seventh section is unclear! Do you mean therapeutic approaches? How many questions in this section? One (q no. 23) or more?

Validity of the findings

RESULTS
L151
I suggest using brackets, for example, (n=187 (49.15 %)).

L154
This sentence “those from Canada and America made up the smallest percentage (n=1,0.3%)” is unclear! Could you correct it? There is only one; I think it should be from Canada or USA!

L155
I suggest spelling out the abbreviation when it is first mentioned in the manuscript.

L159
This are some mistakes in this sentence that should be corrected.

L180
This sentence is unclear! Do you mean that they were ware of the role of PTs in assessing…..?

L182- L184
I suggest rewriting this sentence to be understood clearly.

L145
I see here there were only around 156 responses only! What about the rest of doctors participating in the survey?

L229
This sentence should be rewritten as it is unclear.

Table 2
I suggest writing the percentages in brackets.

Table 3
“Vestibular Conditions referred by medical doctors to physiotherapist (n=27)”…. Do you mean here number of referred cases or number of doctors who referred to PTs? Please, rewrite this sentence clearly.

DISCUSSION
L237
This sentence is repeated: “Less than a quarter of the medical doctors referred cases to physiotherapists.”; I recommend deleting it.

L241
I see that this result reported here is different from that reported in the results section which was talking about the referrals from doctors. Please, correct it.

L252
This paragraph is unclear! The authors talked in the beginning about the need to change the diagnosis/treatments, then they changed the scope of the paragraph going to talking about the effect of exercises in the management of vestibular disorders.

L275
I could not find the results reported here in the results section!

L284
I could not find the results reported here in the results section!

L316
One limitation is the low number of responses. Another big limitation of this study is including doctors who do not have any relation with vestibular disorders.

L319
The online questionnaire was only 10 mins! Do you think it is a long time?

L326
Why especially ENT and general physicians? The most referrals as I see in the results came form those doctors!

Reviewer 3 ·

Basic reporting

For the most part it was very well written. There were some anomalies where I was unsure what you were meaning. For example Line 61 'mushrooming'. I suggest rereading the document and removing or rewording any non academic wording or phrases.

I am also unclear what the paragraph starting on line 252 is trying to achieve. There are many great studies reported, but there is no discussion as to what those other studies add to this current study or how they are related to the findings of this current study. Further synthesis of the literature is needed.

Nice, clean, clear tables and figures.

Experimental design

Experimental question was designed well. The survey was designed well to answer the questions.

Validity of the findings

The findings are briefly discussed. The paper would benefit from discussing the implications of the findings and strategies to improve upon the poor referral to physiotherapists for vestibular disorders. It is not enough to say that physiotherapists are under utilised, but a discussion about the implications of that would provide a greater insight into why these findings matter.

Additional comments

The Discussion should highlight more why it is important for physiotherapists to be involved in the care of people with vestibular disorders, assuming that is what you are trying to achieve? Do patients have better outcomes if they visit a physiotherapist versus a medical doctor? The why physiotherapist and not ENT doctor or other GP to treat this condition is unclear.

---

## Round 0.2 · Minor Revisions

I would like to thank the authors for their hard work in attending to many of the concerns of the three reviewers of the initial submission of the manuscript. While only the first reviewer has provided feedback on this resubmission, I feel that there are still a number of concerns raised by the second and third reviewer that still need further amendments prior to this manuscript being more seriously considered for publication in PeerJ. These include the following comments (with the line numbers relating to the PDF version of the resubmission):

Line 151 and multiple other sentences, potentially also the tables: I disagree with the way one of the reviewers suggested you write these results and percentages. I would suggest you follow this template for the data presented on line 151 and use this throughout the document. The data should be written as (n-187, 49.2%). As suggested here, I also don’t believe there is any benefit from adding in two decimal places, so please report these values in the text as well as in your tables to one decimal place only.

Line 184 – 186: this sentence needs to be rewritten to some extent. Perhaps something like “The remainder of the 21 responses from the medical doctors stated that a lack of confidence (add n and percent values here), lack of experience (add n and percent values here) and insufficient knowledge among physiotherapists (add n and percent values here) were the primary barriers.

Line 252 – 254: you have already stated this on line 246 – 248. Therefore, this sentence can be removed from here as it is redundant.
Line 258 – 261: I suggest this sentence be removed as it doesn’t add anything to the manuscript, rather it reads like a somewhat superficial part of the introduction.

Line 265 – 266: this should read as “A number of research studies have examined the potential for modern physical therapy ……”.
Line 273 – 288: I suggest this sentence be removed as it doesn’t add anything to the manuscript, rather it reads like a somewhat superficial part of the introduction.

Line 356-359: I think you need to expand this a little bit in regards to the limitations of your study. The first concerns the response rate of 30%. The second reflects the fact that somewhere around half of your sample where not necessarily highly active with patients with vestibular disorders, which further reduced the size of your sample. In terms of the generalisability of these results, how similar were the demographic and professional characteristics of your participants to those of the wider Saudi Arabian population of these health professionals?

Overall discussion: I think there also needs to be a bit more presented in the discussion regarding what the primary barriers are that underlie the relatively low prevalence of physiotherapy referrals by medical practitioners for patients with vestibular disorders. For example, does it primarily reflect a lack of knowledge and respect of physiotherapists by medical practitioners in this country, a lack of knowledge of the medical practitioners on the benefits of exercise for this patient group, wider health system challenges etc? Some more discussion of these factors and some potential strategies/recommendations to improve the situation and patient outcome is needed. It would also be great to see some of these recommendations embedded in a relevant theory of behaviour change etc.

Reviewer 1 ·

Basic reporting

Well Reported

Experimental design

Authors have clarified my concerns regarding the design

Validity of the findings

The findings can influence clinical practise in the field of vestibular rehab

Additional comments

The last reply to my question which is copy pasted below should be included in the manuscript

"Thanks a lot for pointing it out. if the medical doctors answered that they did not see the patient with vestibular patients. The questionnaire was terminated after collecting demographic information. "

---

## Round 0.3 · Minor Revisions

I thank the authors for their attendance to most of the previous requests from the reviewers and I. There are just some minor typographical errors that require correction prior to my recommendation that this paper be accepted for publication in PeerJ. These corrections refer to the page numbers in the track changes word manuscript. These include:

Page 2: my request for you to change data presented with 2 decimal places to one decimal place was only with respect to the percentages of your sample who responded in a particular way, not with respect to the chi-square statistic. Therefore, please leave this as “482.476” as such detail can be useful for individuals performing meta analyses. This correction also needs to be applied to all of the other chi-square, odds ratio and confidence interval statistics throughout the manuscript, such as on page 10.

Page six: this revised section needs some of the capitalisation altered and should read as “If the medical doctors answered that they did not see the patient with vestibular patients, the questionnaire was terminated after collecting demographic information.”

Page 17: I suggest some small typographical changes to the sentence in the revised paragraph. Specifically, the second sentence might read better as “The Saudi Arabian doctors may benefit not only from further education into benefits of physical therapy for their patients, but also behaviour change theories and approaches, so that they can better promote the physical therapy services to their patients.

Page 17: the following section in the limitations may also require typographical changes. Specifically, I request the following change “These low response rates and relative equivalence in the medical doctors who are involved or not involved with patients with vestibular disorders, may affect the generalisation of these results to the population of Saudi Arabian doctors.

---

## Round 0.4 · accepted · Accept

Thanks for attending to my requests. I'm now happy to recommend this paper accepted for publication in PeerJ.